# Development of monoclonal antibodies targeting the conserved fragment of hexon protein to detect different serotypes of human adenovirus

Linfan Wu,[1,2,3] Yuhao Lin,[2,3] Juzhen Yin,[4] Changbi Yang,[2,3] Yushan Jiang,[2,3] Linlin Zhai,[2,3] Yuelin Wang,[2,3] Li Zhu,[2,3] Qinghua Wu,[2,3] Bao Zhang,[2,3] Chengsong Wan,[2,3] Wei Zhao,[2,3] Yang Yang,[4] Chenguang Shen,[2,3] Weiwei Xiao[1,2,3]

**ABSTRACT**  Human adenovirus (HAdV) infects the respiratory system, thus posing a threat to health. However, immunodiagnostic reagents for human adenovirus are limited. This study aimed to develop efficient diagnostic reagents based on monoclonal antibodies for diagnosing various human adenovirus infections. Evolutionary and homology analyses of various human adenoviral antigen genes revealed highly conserved antigenic fragments. The prokaryotic expression system was applied to recombinant penton, hexon, and IVa2 conserved fragments of adenovirus, which were injected into BALB/c mice to prepare human adenovirus-specific monoclonal antibodies. Enzyme-linked immunosorbent assay (ELISA), indirect immunofluorescence assay (IFA), and Western blotting were used to determine the immune specificity of the monoclonal antibodies. Indirect ELISA showed that monoclonal antibodies 1F10, 8D3, 4A1, and 9B2 were specifically bound to HAdV-3 and HAdV-55 and revealed high sensitivity and low detection limits for various human adenoviruses. Western blotting showed that 1F10 and 8D3 specifically recognized various human adenovirus types, including HAdV-1, HAdV-2, HAdV-3, HAdV-4, HAdV-5, HAdV-7, HAdV-21, and HAdV-55, and 4A1 specifically recognized HAdV-1, HAdV-2, HAdV-3, HAdV-5, HAdV-7, HAdV-21, and HAdV-55. IFAs showed that 1F10, 8D3, and 4A1 exhibited highly selective localization to A549 cells infected with HAdV-3 and HAdV-55. Finally, two antibody pairs that could detect hexon antigens HAdV-3 and HAdV-55 at low concentrations were developed. The monoclonal antibodies developed in this study show potential for detecting human adenoviruses.

**IMPORTANCE**  In this study, we selected the three most conserved antigenic fragments of human adenovirus to prepare a murine monoclonal antibody for the first time, and human adenovirus antigenic fragments with heretofore unheard of degrees of conservatism were isolated. The three monoclonal antibodies with the ability to recognize human respiratory adenovirus over a broad spectrum were screened by hybridoma and monoclonal antibody preparation. Human adenovirus infections are serious; however, therapeutic drugs and diagnostic reagents are scarce. Thus, to reduce the serious consequences of human viral infections and adenovirus pneumonitis, early diagnosis of infection is required. The present study provides three monoclonal antibodies capable of recognizing a wide range of human adenoviruses, thereby offering guidance for subsequent research and development.

**KEYWORDS**  human adenovirus, hexon protein, monoclonal antibodies, ELISA, Western blot, indirect immunofluorescence assay

H uman adenovirus (HAdV) is a member of the genus *Mastadenovirus* in the family Adenoviridae. HAdV is a non-enveloped icosahedral DNA virus with a diameter

Address correspondence to Chenguang Shen, a124965468@smu.edu.cn, Weiwei Xiao, xweiwei74@126.com, or Yang Yang, yyszth2018@163.com.

Linfan Wu, Yuhao Lin, and Juzhen Yin contributed equally to this article. Author order was determined by the time spent on the article.

The authors declare no conflict of interest.

See the funding table on p. 16.

of approximately 70–90 nm, and the DNA core and protein capsid constitute the viral particle (1). The icosahedral protein capsid of HAdV primarily consists of 240 hexon proteins and 12 fiber attachment proteins associated with 12 penton base proteins involved in recognition and binding to cell receptors (2). The capsid contains four minor proteins (IIIa, VI, VIII, and IX) and six core proteins (V, VII, Mu, TP, IVa2, and proteases) (3). HAdVs have three major capsid antigens: hexon, penton bases, and fibers. These structural antigens contain type-, intertype-, group-specific epitopes and neutralizing epitopes. Adenovirus (AdV) hexon proteins exist as trimers and represent the most abundant proteins on the surface of HAdV, and they are also indicators for diagnosis. Some conserved regions in the anterior and middle segments of the hexon may have good exposure and could contain type-specific or group-specific epitopes with solid antigenicity. Penton proteins play vital roles in AdV adsorption and cell internalization. AdV genome-specific packaging involves a packaging sequence consisting of a series of adenosine/thymidine-rich line repeats called A-repeats (4). IVa2 binds to a specific segment of the viral DNA and exists in the form of a polymer; the DNA probe contains the most critical A repeat sequence for the genome capsid (5, 6), suggesting that the interaction between IVa2 and the packaging sequence is crucial in initiating the AdV assembly process.

Furthermore, more than 114 genotypes of HAdVs (http://hadvwg.gmu.edu/) have been recognized and divided into seven species (A–G) (7), including HAdV-3, HAdV-7, HAdV-11, HAdV-14, HAdV-21, and HAdV-55 of species B; HAdV-1, HAdV-2, HAdV-5, and HAdV-6 of species C; and HAdV-4 of species E, which are primarily associated with respiratory tract infection (8). Notably, HAdV-3, HAdV-4, HAdV-5, HAdV-7, HAdV-11, HAdV-14, and HAdV-55 are the main AdVs associated with AdV outbreaks and epidemics (9). The different HAdVs are associated with unique diseases (10); species A causes gastrointestinal tract infections, and species B causes acute respiratory illness and kidney infections. Species C is associated with respiratory tract and lymphoid tissue infections, species D and E cause keratitis and conjunctivitis, and species F causes diarrhea in infants and young children. HAdV can infect people of all ages; the AdV-susceptible population includes infants, children, military recruits, and immunocompromised patients (11). In addition, the prevalence of HAdVs is closely related to geographical location (12). For example, the risk of transmission of HAdV infection is increased among crowded individuals in a closed space; therefore, the transmission of HAdVs is clustered and explosive (13). Respiratory failure and severe adenovirus pneumonia may occur in infants and people with immune deficiency or low immunity (10). Furthermore, HAdV infection can lead to acute respiratory distress syndrome and death (14). This study revealed that 5%–10% of respiratory tract infections in children and 1%–7% of adult respiratory tract infections are caused by AdVs; >20% of neonatal or child pneumonia is caused by AdVs, and the severe AdV pneumonia mortality rate is >50%. Two related studies in children with adeno-associated virus 2 (AAV2) infection in the United Kingdom reported the presence of unexplained liver damage or hepatitis; this indicates that AAV2 may cause liver damage in genetically susceptible children through specific immune mechanisms. Notably, AAV2 requires a helper virus, most commonly HAdV or herpes virus, to replicate in the liver and infect the hepatocytes (15, 16). Early diagnosis and treatment of viral infections can markedly shorten the course of the disease and reduce the severity and mortality caused by viral infections (17–20). Therefore, early diagnosis of adenovirus infections is imperative.

Virus isolation and culture, immunological detection, molecular biology detection, and viral morphology observation are classic clinical and laboratory testing methods. Although virus isolation is the most classic isolation and identification method (21), it is time consuming, laborious, and highly subjective and requires biosafety laboratories and professional equipment. Therefore, it is unsuitable for large-scale, rapid, and accurate clinical testing requirements. Molecular biology detection reagents rely on comparing and designing specific primers, and methods based on the molecular biology detection of specific nucleic acid sequences include polymerase chain reaction (PCR)

series methods, such as ordinary PCR, nested PCR, real-time fluorescent quantitative PCR, and PCR-enzyme-linked immunosorbent assay (ELISA). PCR amplification technology is currently the most extensively used molecular diagnostic method (22), and it can identify the serotype of AdV within a few hours; however, there are specific requirements for this approach. For example, the detection requirements are highly dependent on professional technicians and equipment; additionally, there is a risk of contamination and false positives (23).

Furthermore, gene sequencing and biochips are used in molecular biology techniques. Comparatively, gene sequencing can determine the AdV type more rapidly; however, the operation cost is high. Chip technology has the advantages of high-throughput, reasonable specificity, high sensitivity, and miniaturization; however, the high requirements and technical costs limit its application. These tests have certain limitations, and meeting the needs of many practical and rapid diagnoses is challenging. Immunological detection requires antibodies that specifically react with various HAdV types. Immunological antigen-antibody reactions can be detected using immunochemical techniques, including the immunofluorescence method, colloidal gold method, and ELISA, which generate results within a short period; however, they exhibit different sensitivities based on the specific antibody quality and design. Therefore, this study aimed to develop novel monoclonal antibodies (mAbs) that recognize an extensive range of HAdV serotypes.

Köhler and Milstein pioneered the B lymphocyte hybridoma cell and mAb technology in 1975. A mAb is a universal binding molecule that is an indispensable tool in research, diagnosis, and therapy (24, 25), and it has extensive application prospects. The fundamental principle is that spleen cells from mice immunized with specific antigens are fused with mouse myeloma cells to generate hybridomas, which represent clones resulting from the fusion of B cells; they are capable of indefinite myeloma cell proliferation and are involved in the synthesis and secretion of specific antibodies inherited from immune B cells. mAb technology is based on the principle of clonal expansion of lymphocytes, which involves selective screening of monoclonal cells with specific antigen specificity to obtain a highly purified single type of antibody. However, few mAbs can detect multiple serotypes of AdVs (26–28), and the development of broad-spectrum mAbs for AdV diagnosis with better performance is necessary. In addition, no therapeutic drugs are currently available for effectively treating HAdV infection. Despite cases indicating the efficacy of cidofovir, its safety and effectiveness must be further elucidated (29). Furthermore, preventive HAdV vaccines are not used globally; however, approved live AdV types 4 and 7 vaccines are exclusive to the US military (30).

In conclusion, rapid diagnosis of human adenovirus is crucial for preventing HAdV infection and controlling virus transmission, especially for preventing and treating severe pneumonia caused by AdVs. This study developed specific mAbs that can detect HAdV infection and determine the severity of HAdV infection, representing the premise for developing rapid diagnostic reagents and therapeutic drugs for respiratory adenoviral infection treatment.

## RESULTS

### Molecular evolution, homology, and hydrophilicity analysis of adenovirus genes

Representative gene sequences of the capsid, secondary, and core proteins from HAdV-1, HAdV-2, HAdV-3, HAdV-4, HAdV-5, HAdV-7, HAdV-11, HAdV-21, and HAdV-55 were selected for molecular phylogenetic tree construction and homology analyses (Fig. 1A). The result indicated that the gene fragments of penton protein 418–586, hexon protein 530–983, and IVa2 protein 66–446 were the most conserved gene segments, suggesting that these antigens can induce broad-spectrum detection antibodies against various HAdVs (Fig. 1B). The hydrophilic sequences of the three antigen proteins were analyzed; the three polypeptide sequences were primarily located below the 0 point and were primarily hydrophilic. The grand average hydropathicity (GRAVY) of penton, hexon, and

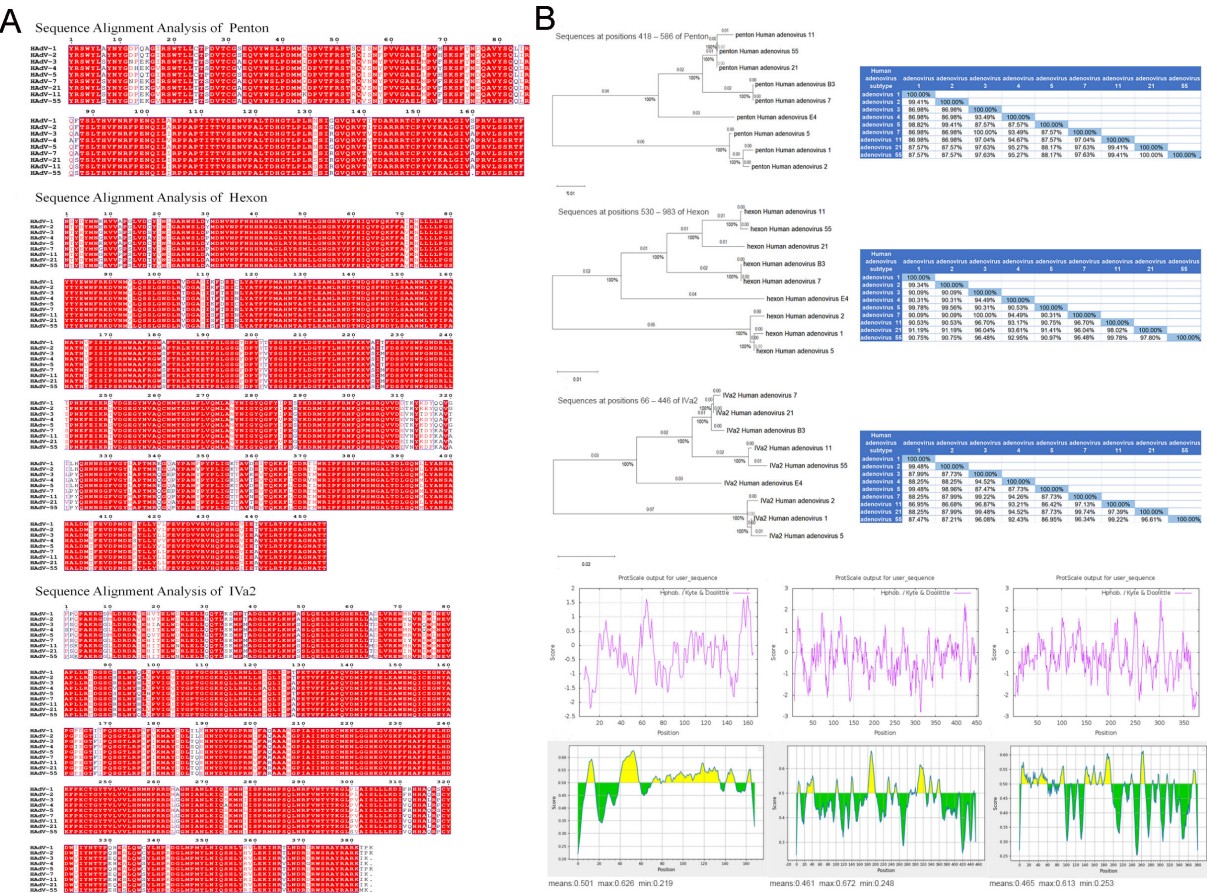

**FIG 1** Sequence alignment of three conserved antigen fragments. Molecular evolution, homology, hydrophilicity, and B cell epitope prediction analyses of the three conserved HAdV proteins. (A) Sequence alignment of penton conserved antigen fragments, hexon conserved antigen fragments, and IVa2 conserved antigen fragments. (B) Molecular evolutionary tree, homology analysis, and B cell epitope prediction of penton protein, hexon protein, and IVa2 protein.

IVa2 were −0.327, −0.292, and −0.331, respectively, indicating that the proteins could be stably maintained. Furthermore, several antigenic determinants were detected in the three peptides by predicting the linear epitopes of B cells.

## Prokaryotic expression and purification of adenovirus conserved protein fragment, establishment of monoclonal cell line, and purification of mAbs

The recombinant plasmids for the HAdV 7E penton, hexon, and IVa2 were transformed into *Escherichia coli* BL21 (DE3) cells, expressed, and purified (Fig. S1). After repeated immunization of BALB/c mice with the purified recombinant penton, hexon, and IVa2 proteins, immune serum titers of the three recombinant proteins were detected. After three booster immunizations, the highest immune serum titer specific to the penton protein was 400, which indicated poor immunogenicity; the highest immune serum titer against the hexon protein was 204,800, and the highest immune serum titer against IVa2 protein was 102,400 (Fig. S2). Splenocytes from mice with the highest immune serum titer of the recombinant hexon and IVa2 protein were fused with SP2/0 cells to generate five strong positive monoclonal cell lines against the HAdV hexon protein and one strong positive monoclonal cell line against the HAdV IVa2 protein. The antibodies were produced and purified (Fig. S3).

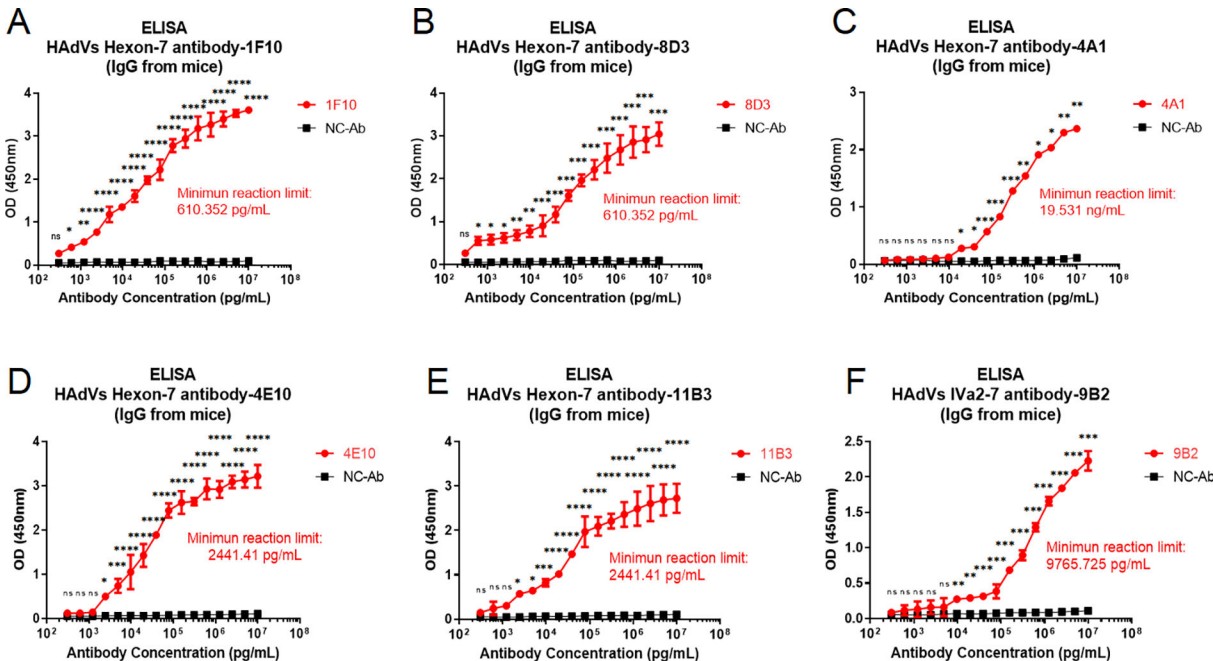

**FIG 2** Antigen binding activities of mAbs for hexon-7 of HAdVs or IVa2-7 protein of HAdVs. (A) Reactivity of 1F10 with HAdVs hexon-7 protein. (B) Reactivity of 8D3 with HAdVs hexon-7 protein. (C) Reactivity of 4A1 with HAdVs hexon-7 protein. (D) Reactivity of 4E10 with HAdVs hexon-7 protein. (E) Reactivity of 11B3 with HAdVs hexon-7 protein. (F) Reactivity of 9B2 with HAdV IVa2-7 protein. ns, $P < 0.1234$; *$P < 0.0332$; **$P < 0.0021$; ***$P < 0.0002$; ****$P < 0.0001$.

## Binding specificity of mAbs to the corresponding recombinant adenovirus protein or human adenovirus by ELISA

The binding reactivity of five mAbs to the hexon protein of HAdV and one mAb to the IVa2 protein of HAdV was detected using indirect ELISA, and the results showed that all antibodies had good binding reactivity. The minimum reaction limits of the 1F10, 8D3, 4A1, 4E10, and 11B3 antibodies against the hexon protein antigen of HAdV were 610.352 pg/mL, 610.352 pg/mL, 19.531 ng/mL, 2,441.410 pg/mL, and 2,441.410 pg/mL, respectively (Fig. 2A through E). The minimum reaction limit of the 9B2 antibody to the IVa2 protein antigen of HAdV was 9,765.725 pg/mL (Fig. 2F). Furthermore, antibodies 1F10, 8D3, 4A1, and 9B2 showed good binding activity to HAdV-3 and HAdV-55, with a minimum reaction limit for HAdV-3 of 625, 1,250, 2,500, and 1,250 ng/mL, respectively (Fig. 3A through D) and minimum reaction limits for HAdV-55 of 625, 2,500, 2,500, and 2,500 ng/mL, respectively (Fig. 3E through H). Considering that the epitope recognized by the antibody may not be exposed on the surface of the antigen, we attempted to denature the virus by lysis to improve antibody reactivity. When using the lysate, the binding activities of the 1F10, 8D3, and 4A1 antibodies for the virus improved to a certain extent. For example, the minimum reaction limit of 1F10 for HAdV-3 and HAdV-55 decreased from 625 ng/mL to 312.5 ng/mL (Fig. 3I) and from 625 ng/mL to 312.5 ng/mL (Fig. 3J), respectively, while that of 8D3 for HAdV-3 and HAdV-55 decreased from 1,250 ng/mL to 625 ng/mL (Fig. 3K) and 2,500 ng/mL to 625 ng/mL (Fig. 3L), respectively. Furthermore, the binding activity of antibody 4A1 for HAdV-3 and HAdV-5 showed a similar trend (Fig. 3M and N), while that of mAb 9B2 to HAdV-3- and HAdV-55-diluted lysate was not substantially improved.

## Diagnosis of HAdV infection using HAdV mAb-based immunoblotting

Western blotting showed that mAbs 1F10 and 8D3 specifically recognized HAdV-1, HAdV-2, HAdV-3, HAdV-4, HAdV-5, HAdV-7, HAdV-21, and HAdV-55 (Fig. 4A and B) and mAb 4A1 specifically recognized HAdV-1, HAdV-2, HAdV-3 HAdV-5, HAdV-7, HAdV-21, and HAdV-55 (Fig. 4C). A band of approximately 120 kDa was observed. According to the

relative molecular mass of the HAdV structural protein, it was deduced that the protein at 120 kDa was the hexon subunit. In this study, AdV DNA-binding protein (DBP) was selected as a positive control. DBP is the product E2A of AdV E2 region gene expression (DNA-binding protein), and GAPDH was used as a standardized internal reference (Fig. 4D). We conducted immunoblot experiments using the non-denaturing adenovirus as a reactant; notably, HAdV antigens could not be detected under natural conditions.

## Indirect immunofluorescence detects the localization of mAbs on cells before and after virus replication

HAdV-3 and HAdV-55 viral dilutions were added to the cells in good growth conditions and incubated to infect the cells; A549 cells without viral infection were used as control. The recognition and localization of antibodies to cells before and after viral infection were observed using fluorescence microscopy; the immunofluorescence results show that the mAbs 1F10, 8D3, and 4A1 were precisely localized to the cells after infection with HAdV-3 and HAdV-55. Conversely, the mAbs did not localize to uninfected cells (Fig. 5A through C). The localization of mAbs showed a statistically significant difference (Fig. 5D), indicating that the mAbs were stimulated after replication of the virus.

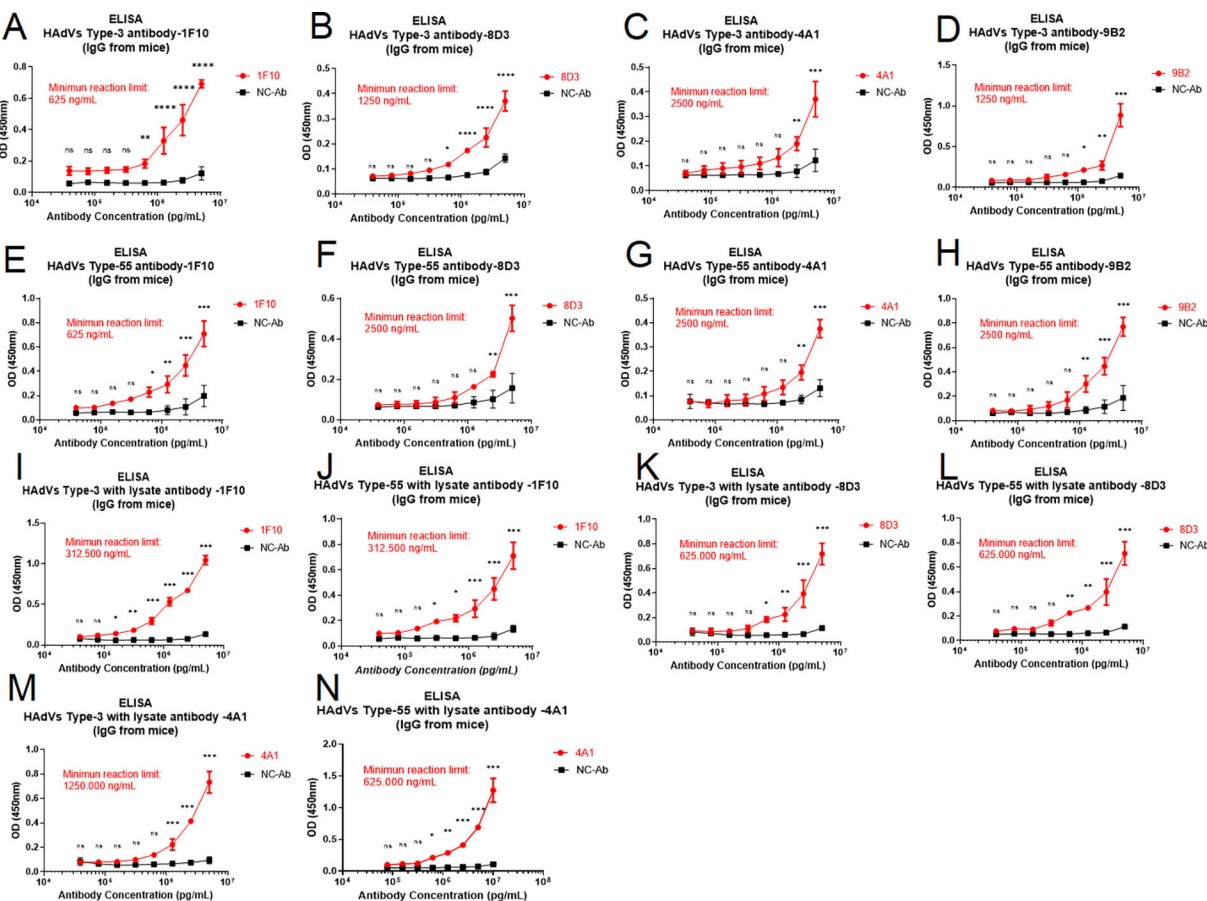

**FIG 3** HAdV-3 and HAdV-55 detection activities of three mAbs for hexon-7 protein; HAdV-3 and HAdV-55 detection activities of one mAb for hexon-7 IVa2; binding activities of mAb for virus treated with the lysate. (A) Reactivity of 1F10 with HAdV-3. (B) Reactivity of 8D3 with HAdV-3. (C) Reactivity of 4A1 with HAdV-3. (D) Reactivity of 9B2 with HAdV-3. (E) Reactivity of 1F10 with HAdV-55. (F) Reactivity of 8D3 with HAdV-55. (G) Reactivity of 4A1 with HAdV-55. (H) Reactivity of 9B2 with HAdV-55. (I) Reactivity of 1F10 with HAdV-3 treated with the lysate. (J) Reactivity of 1F10 with HAdV-55 treated with the lysate. (K) Reactivity of 8D3 with HAdV-3 treated with the lysate. (L) Reactivity of 8D3 with HAdV-55 treated with the lysate. (M) Reactivity of 4A1 with HAdV-3 treated with the lysate. (N) Reactivity of 4A1 with HAdV-55 treated with the lysate. ns, $P < 0.1234$; *$P < 0.0332$; **$P < 0.0021$; ***$P < 0.0002$; ****$P < 0.0001$.

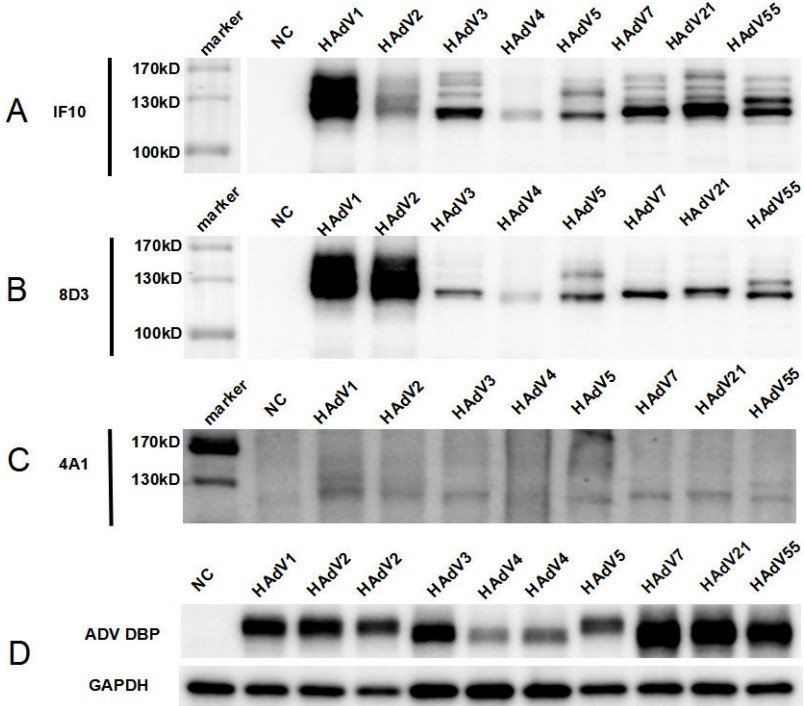

FIG 4 Western blot analysis of immunological specificity of the mAbs to the respiratory HAdV. (A) Immunological specificity of antibody 1F10 to HAdV-1, HAdV-2, HAdV-3, HAdV-4, HAdV-5, HAdV-7, HAdV-21, and HAdV-55. (B) Immunological specificity of antibody 8D3 to HAdV-1, HAdV-2, HAdV-3, HAdV-4, HAdV-5, HAdV-7, HAdV-21, and HAdV-55. (C) Immunological specificity of antibody 4A1 to HAdV-1, HAdV-2, HAdV-3, HAdV-5, HAdV-7, HAdV-21, and HAdV-55. (D) Immunological specificity of ADV DBP and GAPDH to HAdV-1, HAdV-2, HAdV-3, HAdV-4, HAdV-5, HAdV-7, HAdV-21, and HAdV-55. (Fig. 4A and B add the gel spliced for labeling sample size.)

## Detection of human adenovirus antigen using double-antibody sandwich ELISA

Double-antibody sandwich ELISA results revealed that the hexon antigen could be detected at a minimum concentration of 312.5 ng/mL when the capture antibody 1F10 and the detection antibody 8D3 were employed (Fig. 6A). The minimum detection limit for HAdV-3 was 352 PFU (Fig. 6B), and HAdV-55 demonstrated a minimum detection limit of 516 PFU (Fig. 6C). In contrast, when the capture antibody 4A1 and detection antibody 1F10 were used, the minimum detection limit was 250 ng/mL for the hexon antigen (Fig. 6D), 704 PFU for HAdV-3 (Fig. 6E), and 516 PFU for HAdV-55 (Fig. 6F). However, the remaining paired antibodies exhibited poor binding specificity for the viral antigens. Simultaneously, we determined the Threshold cycle (Ct) value of the virus using qPCR, and it was limited to a maximum of 35. A Ct value that exceeded 35 indicated a low virus load and no risk of infection. In this experiment, when HAdV-3 was diluted 1,250–6,250 times and HAdV-55 was diluted 50–250 times, the Ct values exceeded 35 (Fig. 6G). Therefore, we could estimate that the minimum concentration of HAdV-3 detected using qPCR was 90 PFU and that of HAdV-55 was 413 PFU.

## Sequence analysis of heavy-chain and light-chain gene variable region of mAb

The sequencing results were analyzed on the VBASE2 website to obtain the CDR sequences of the heavy and light chains of mAbs 1F10, 8D3, and 4A1 (Fig. S4).

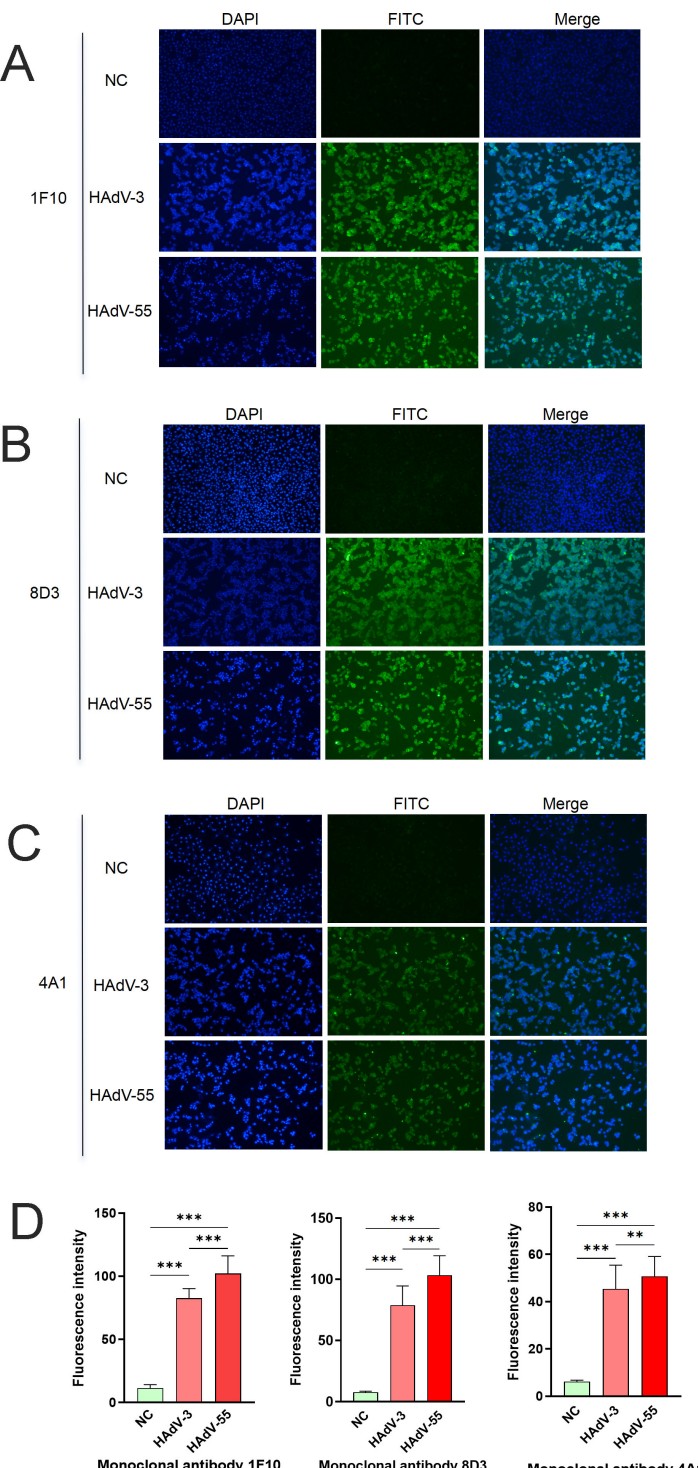

**FIG 5** Indirect immunofluorescence detection of the localization of mAbs 1F10, 8D3, and 4A1 on cells before and after replication of HAdV-3 and HAdV-55. (A) Indirect immunofluorescence was used to detect the localization of 1F10. (B) Indirect immunofluorescence was used to detect the localization of 8D3. (C) Indirect immunofluorescence was used to detect the localization of 4A1. (D) Fluorescence intensity analysis of mAbs; 10× objective lens; green is the fluorescence expressed by the Fluorescein Isothiocyanate (FITC) channel antibody, and blue is the DAPI nuclear stain. **$P < 0.0021$; ***$P < 0.0002$.

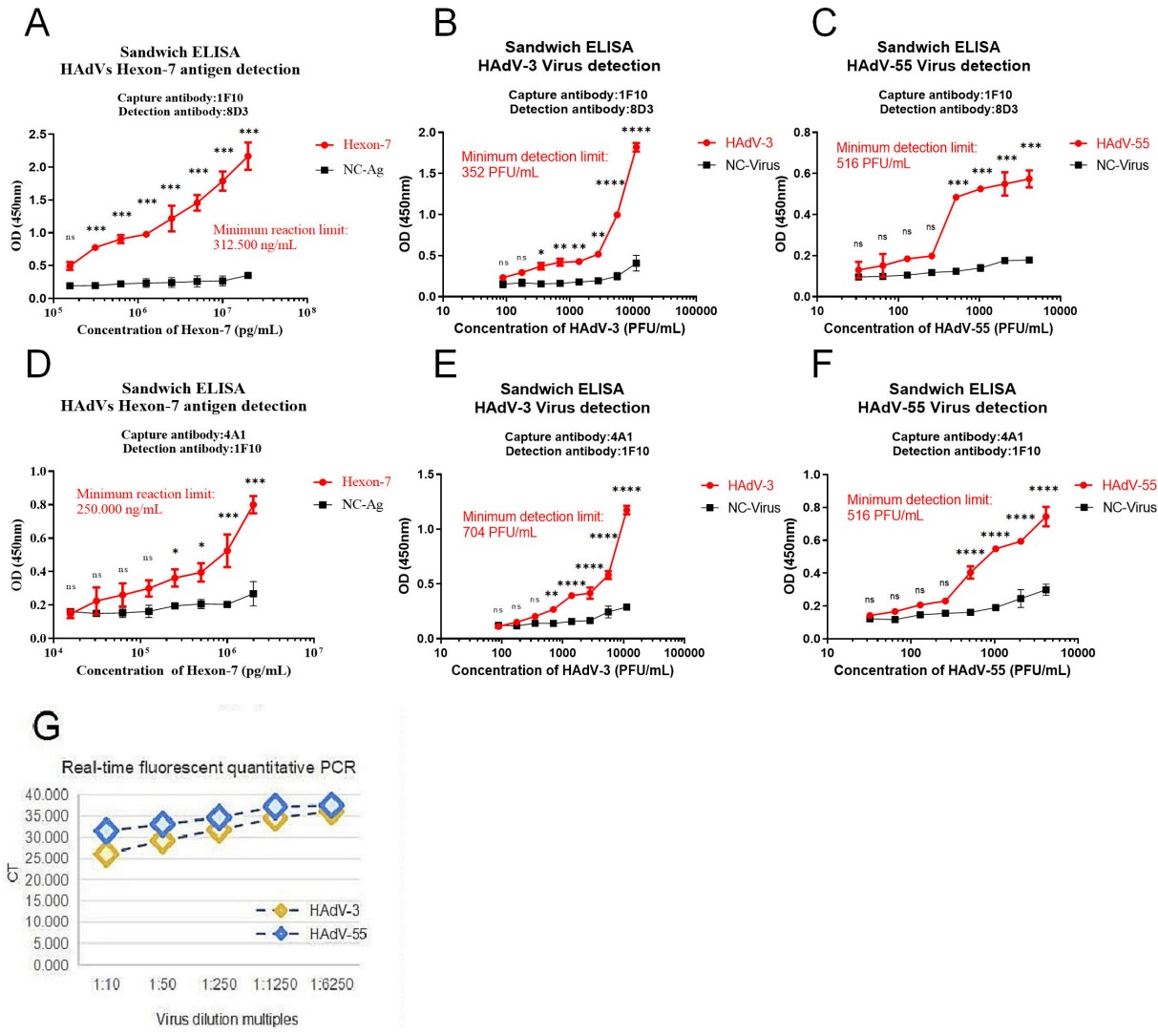

**FIG 6** Hexon antigen, HAdV-3, and HAdV-55 were detected using double antibody sandwich ELISA; HAdV-3 and HAdV-55 were detected using qPCR. (A) Detection of hexon antigen using the 1F10-8D3 sandwich ELISA. (B) Detection of HAdV-3 using the 1F10-8D3 sandwich ELISA. (C) Detection of HAdV-55 using the 1F10-8D3 sandwich ELISA. (D) Detection of hexon antigen using the 4A1-1F10 sandwich ELISA. (E) Detection of HAdV-3 using the 4A1-1F10 sandwich ELISA. (F) Detection of HAdV-55 using the 4A1-1F10 sandwich ELISA. (G) The Ct values of HAdV-3 and HAdV-55 were detected by qPCR ($R^2 = 0.994$). ns, $P < 0.1234$; *$P < 0.0332$; **$P < 0.0021$; ***$P < 0.0002$; ****$P < 0.0001$.

## DISCUSSION

HAdV is a common pathogen that causes acute respiratory infectious diseases, and it can infect various tissues, causing respiratory infection, gastrointestinal tract infection, and conjunctiva. HAdV shows a global prevalence with no apparent seasonality. Multiple HAdVs can be associated with an infection; for example, >10 HAdVs can cause severe respiratory infections in humans. Although HAdV is a DNA virus whose nucleic acid mutation is not as rapid as influenza or HIV, it is significantly complex and diverse. HAdVs have strong adaptability to the environment, and mutated strains constantly appear. Additionally, new serotypes of HAdV have been discovered, and cases of mixed infections or cross-infections have been reported. Therefore, there is an urgent need to develop a new and rapid diagnostic method for detecting HAdVs and preventing and controlling infection. The neutralization test is an immunological test used to identify pathogens and determine their serotypes; however, studies on HAdV-neutralizing antibodies, especially broad-spectrum neutralizing antibodies, are limited. The mAbs

1F10, 8D3, and 4A1 developed in this study were used for HAdV neutralization reactions; however, satisfactory results were not obtained. Notably, only a few strains of HAdV type-specific neutralizing antibodies have been reported. For example, the neutralizing antibody 9C12 that explicitly targets the HAdV type 5 hexon protein was reported in 2004 (31); mouse neutralizing antibody 1B6 that explicitly targets the HAdV type three hexon protein was reported in 2013 (32); the mouse-derived antibody MN4b that explicitly targets the hexon protein of HAdV type 4 was reported in 2018 (33); and mouse antibody 10G12 that explicitly targets the hexon protein of HAdV type 7 was reported in 2019 (34). The human neutralizing antibody 3-3E specifically targets the hexon protein of human adenovirus type 7 (35). Additionally, murine antibodies 3F11 and 3D8 extensively neutralize HAdV-7, HAdV-11, and HAdV-55 of species B (36). These mAbs have high specificity and neutralization activity but do not provide the corresponding gene sequence; therefore, these antibodies failed to express the corresponding mAbs as a contrast. However, HAdV-3 and HAdV-55 were detected using qPCR and were compared with the HAdVs detected using ELISA.

This study selected representative HAdV penton, hexon, and IVa2 proteins, and gene sequence alignment was initially performed. A molecular evolutionary tree was constructed, and the homology of the HAdV sequence was analyzed to obtain conserved proteins. Three conserved antigen fragments are expressed by the prokaryotic gene system, which has the advantages of simple cultivation, high transformation rate, and high efficiency expression. However, inclusion bodies were easily formed when *E. coli* was highly expressed; this phenomenon is primarily because *E. coli* does not exhibit post-translational modification and glycosylation functions and does not possess the mechanism of effectively secreting foreign proteins outside the cell and forming disulfide bonds (37). Although inclusion bodies are non-folded aggregates with no biological activity, the protein can be restored after denaturation and renaturation of inclusion bodies (38). After achieving optimal refolding conditions, the proteins were successfully refolded using urea gradient dialysis. However, only the recombinant hexon and IVa2 proteins showed good antigenicity, and the mice were successfully immunized. The penton protein has a low molecular weight, may not form a suitable epitope, and fails to initiate a strong immune response in immunized mice. Five specific mAbs against the HAdV exon protein and one specific mAb against the HAdV IVa2 protein were obtained from purified ascitic fluid. Indirect ELISA showed that mAbs 1F10, 8D3, and 4A1 could detect HAdV-3 and HAdV-55. In addition, they were more sensitive to HAdV-3 and HAdV-55 when treated with lysates. Western blotting showed that the mAbs 1F10 and 8D3 could specifically recognize HAdV-1, HAdV-2, HAdV-3, HAdV-4, HAdV-5, HAdV-7, HAdV-21, and HAdV-55, and 4A1 could specifically recognize HAdV-1, HAdV-2, HAdV-3, HAdV-5, HAdV-7, HAdV-21, and HAdV-55; however, they could not recognize HAdVs under natural conditions, indicating that mAbs 1F10, 8D3, and 4A1 recognize linear epitopes. Furthermore, indirect immunofluorescence assay revealed that these mAbs precisely identified HAdV-infected cells. Simultaneously, we used a double-antibody sandwich pairing test, such as 1F10-8D3 paired antibody and 4A1-1F10 paired antibody, which can specifically detect low concentrations of the hexon antigen, HAdV-3, and HAdV-55.

Western blotting revealed that these mAbs could detect different serotypes of adenovirus; however, only the binding activity of mAbs against HAdV-1, HAdV-2, HAdV-3, HAdV-4, HAdV-5, HAdV-7, HAdV-21, and HAdV-55 was detected, and all AdV serotypes could not be elucidated. Whether the mAbs used in this study can detect all AdV serotypes should be further studied. Notably, these antibodies could not confirm an ongoing infection or establish a direct link with the symptoms because they are non-type specific; therefore, they can not be employed to determine the infecting HAdV type. The antibodies developed in this study have not been validated in clinical samples of adenoviral infections. Although the sensitivity of detection of ELISA is lower than qPCR, ELISA is simpler to operate, does not require professional precision instruments and equipment or skilled personnel to operate, enables a faster detection time at low

cost, and is more suitable for large-scale detection. Furthermore, these antibodies should be used in future research to detect more HAdV types, and colloidal gold rapid detection kits should be developed to rapidly identify various HAdVs.

## Conclusion

Specific mAbs 1F10, 8D3, and 4A1 against the HAdV hexon protein antigen were successfully prepared in this study. After preliminary confirmation, specific HAdV antibodies were used to confirm whether the individuals were infected with HAdV. Furthermore, such antibodies could be of interest in describing the natural history of HAdV infection and understanding how HAdV concentrations in body fluids correlate with the protection or severity of infection. Therefore, this study has laid a robust foundation for human epidemiology and future research on rapid, sensitive, and specific diagnostic reagents for HAdVs.

## MATERIALS AND METHODS

### Primary materials

HAdV-3, HAdV-7, HAdV-21, and HAdV-55 of species B; HAdV-1, HAdV-2, and HAdV-5 of species C; and HAdV-4 of species E were isolated from the Third People's Hospital of Shenzhen; expression plasmids for penton (human adenovirus 7), hexon (human adenovirus 7), and IVa2 (human adenovirus 7) were synthesized by Sango Biotech; *Escherichia coli* BL21 (DE3) and *Escherichia coli* DH5α were preserved in our laboratory; SPF grade 6- to 8-week-old female BALB/c mice were purchased from Guangdong Medical Experimental Animal Center, and SP2/0 and A549 cells were cultured and preserved in our laboratory.

### Primary reagents

Urea, arginine, and glycerol were purchased from MACKLIN; Tris and ethylenediamine-tetraacetic acid (EDTA) were purchased from Biofroxx; sodium chloride was purchased from Guangdong Guanghua Sci-Tech Co., Ltd.; dithiothreose alcohol, 3, 3', 5, 5' -tetramethylbenzidine (TMB) substrate for ELISA, ELISA stopping solution, glue-activated horseradish peroxidase labeling kit, and Triton X-100 were purchased from Solarbio; Freund's complete adjuvant, Freund's incomplete adjuvant, HAT, and HT were purchased from Sigma; goat anti-mouse IgG (H&L) horseradish peroxidase (HRP) was purchased from Sango Biotech; fetal bovine serum and RPMI 1640 medium were purchased from ThermoFisher Scientific; PBS was purchased from GBCBIO Technologies; saturated ammonium sulfate solution was purchased from Leagene; PAGE gel quick preparation kit was purchased from ATGene; and non-reducing loading buffer and reducing loading buffer were purchased from GBCBIO Technologies, Inc. Confocal dishes were purchased from NEST Biotechnology Co., and 4% paraformaldehyde, goat serum, Hoechst 33342 staining solution, and anti-fluorescence quenching sealer were purchased from Beyotime. Fluorescein (FITC)-conjugated goat anti-mouse IgG (H + L) was purchased from Proteintech.

### Conservation analysis of human adenovirus antigen gene

We downloaded the full-length nucleic acid gene sequences of the capsid protein, secondary protein, and core protein of human respiratory AdV type 1, type 2, and type 5 of species C; type 3, type 7, type 11, type 21, and type 55 of species B; and type 4 of adenovirus species E from GenBank (http://www.ncbi.nlm.nih.gov/genbank/) (Table 1). Next, using ClustalW in Mega software (version 11.0), the downloaded sequences were aligned and compared to delete redundant sequences, and the results were exported using ESPript 3.0 (ESPript 3 .x/ENDscript2.x). Subsequently, the phylogenetic analysis function of the Mega 11.0 software was used to construct the molecular phylogenetic

**TABLE 1** Human respiratory adenovirus types and accession numbers obtained from GenBank

| Type | Penton accession number | Hexon accession number | IVa2 accession number |
|---|---|---|---|
| Adenovirus 1 | AYP21301 | BAG48778 | AYP21296 |
| Adenovirus 2 | CAC67483 | BAG48779 | AP_000165 |
| Adenovirus 3 | CAA82622 | BAG48780 | ANQ44524 |
| Adenovirus 4 | ANQ44490 | BAG48781 | ANQ44484 |
| Adenovirus 5 | AAA42519 | BAG48782 | AP_000201 |
| Adenovirus 7 | AEC11865 | BAG48784 | QEQ50289 |
| Adenovirus 11 | QZA82864 | BAG48788 | QZA82858 |
| Adenovirus 21 | QXM26821 | BAG48798 | QXM26808 |
| Adenovirus 55 | QPD79298 | QPD79303 | AXN73958 |

tree and analyze the homology of the downloaded sequences using the neighbor-joining method and JTT model (https://www.ebi.ac.uk/Tools/msa/clustalo/). Based on the alignment results, we selected three polypeptide fragments with the highest homology, which were penton, hexon, and IVa2 gene fragments. The hydrophilicity of the proteins was analyzed using the Expasy website (https://web.expasy.org/protscale/). Finally, we chose IEDB tools (http://tools.iedb.org/main/bcell/) to predict B cell linear epitopes and easily obtain expressed conserved proteins.

## Prokaryotic expression and purification of recombinant penton protein, hexon protein, and IVa2 protein

Three plasmids, penton HAdV-7, hexon HAdV-7, and IVa2 HAdV-7, were constructed using pET-28α (+) as a vector. The penton HAdV-7 expression fragment composed of amino acids was found at location 418–586, and the relative molecular mass was approximately 19,270 Da; hexon HAdV-7 expression fragment was composed of amino acids and found at location 530–983, and the relative molecular mass was approximately 51,905 Da; and IVa2 HAdV-7 expression fragment was composed of amino acids and found at location 66–446, and its relative molecular mass was approximately 43,467 Da. These recombinant expression plasmids were transformed into *E. coli* BL21 (DE3) cells and cultured overnight in Luria-Bertani (LB) agar culture dishes containing the corresponding antibiotics. Next, a single colony was selected and inoculated into an LB sterile liquid medium (containing 50 mg/L kanamycin). After generating several activation cultures, isopropyl-beta-D-thiogalactopyranoside was added to induce expression for 5 h when the $OD_{600}$ of the bacterial solution reached 0.4–0.6. After centrifugation to collect the bacteria, an appropriate amount of lysozyme was added and allowed to stand on ice for 30 min. An ultrasonic wave apparatus was used to crush the cells completely. The above preliminary experiments showed that the three recombinant proteins were expressed as inactive inclusion bodies, and a His-tagged protein agarose high-speed purification resin was used for purification. After the purified inclusion bodies were fully dissolved in 8 mol/L urea solution, the metaproteins were refolded using urea gradient dialysis. The effects of expression, purification, and renaturation were detected using sodium dodecyl sulfate-polyacrylamide gel electrophoresis (SDS-PAGE).

## Mouse immunization

Four to five female BALB/c mice, aged 6–8 weeks, were selected and immunized with purified adenovirus recombinant proteins; purified penton, hexon, and IVa2 proteins were used to immunize mice in different groups. Antigens were emulsified with an equal volume of Freund's complete adjuvant and injected subcutaneously into BALB/c mice. Freund's incomplete adjuvant was used to enhance immune response. All mice were boosted with three 2-week interval immunizations with 50-μg antigen. Next, the orbital blood of the mice was collected, and the serum titer was detected 12 days after the third immunization using indirect ELISA. Before the first immunization, serum was

obtained from the orbital blood of mice as a negative control. When the immune titer was >100,000, the mouse with the highest titer was selected for antibody development.

## Indirect ELISA

The antigen protein was diluted to 1 µg/mL solution with 1× phosphate-buffered saline (PBS), and 100 µL was added to each well of the 96-well microplates. After incubation at 37℃ for 2 h, the plate was washed with 1× PBS containing 0.05% Tween 20 (PBST) three times and incubated with 200-µL 5% skimmed milk powder solution containing 1× PBS for 2 h at 37℃ or overnight at 4℃. The mouse serum incubated at 56℃ for 30 min was serially diluted, and the antibody sample dilutions ranged from 1:200 to 1:102,400. Subsequently, 100 µL of the samples was added to each well of the plate. After washing, 100 µL of goat anti-mouse IgG (H&L) HRP and 5% skimmed milk powder solution was added to each well and incubated at 37℃ for 30 min. After washing, 100 µL of TMB substrate for ELISA was added and incubated at 37℃ for 10–30 min in a dark room, and 50 µL of $H_2SO_4$ solution was added as a stop solution to terminate the reaction, and the $OD_{450}$ was detected within 15 min. Test wells were considered positive when the $OD_{450}$ value of the test wells was >2.1 times that of the negative control wells.

## Cell fusion, positive clone selection, and subclone amplifications

Sufficient SP2/0 cells were prepared in advance, and mouse splenocytes were collected; SP2/0 cells and splenocytes were mixed, and hybridoma cells were cultured in a 96-well cell plate. After 1 week, the medium was changed, and an indirect ELISA was performed to detect the antibody titer in the cellular supernatant. Next, 50-µL cell supernatant was used as the detection antibody to screen the positive cell clones using the detection method described above. A limited dilution method was used to obtain the single-cell subclones, and each subclone was amplified and cultured.

## Ascites preparation and purification

BALB/c mice were intraperitoneally injected with 500 µL of Freund's incomplete adjuvant or liquid paraffin. Next, the expanded monoclonal cell line was injected into the abdominal cavity of the mice after 1 week to produce antibodies. Additionally, a large volume of ascites that contain crude antibodies and large amounts of fat, tumor cells, blood cells, and other impurities were produced in the abdominal cavity of the mice after 1 week. After the ascites were diluted and centrifuged, an equal volume of saturated ammonium sulfate solution was added to precipitate antibodies. After centrifugation, the precipitate was retained, and 1× PBS was used to dissolve the precipitate. Next, the supernatant was collected, and the impurities were removed using 0.22-µm membrane filtration. Filtrates containing antibodies were collected for affinity chromatography purification. Dialysis with 1× PBS was performed, and the buffer was replaced to obtain high-quality mAbs. The purity of the antibodies was detected using SDS-PAGE. Finally, the mAbs were stored at –80℃.

## Titer determination of mAbs

(i) Coating antigen protein, antibody (detection antibody), and mAb titers were detected using indirect ELISA. (ii) The inactivated HAdV-3 and HAdV-55 antigens were used as coating antigens, and indirect ELISA was used to detect the binding titer of antibodies to AdVs. (iii) HAdV-3 and HAdV-55 were diluted with lysates and boiled in a metal bath at 100℃ for 30 min. The composition of 100 mL of lysate is as follows: 20 mL of 500 mmol/L Tris-HCl (pH 8.0); 25 mL of 100 mmol/L EDTA; 10 mL of 5 mol/L NaCl; 1 g of SDS powder; and deionized water to a volume of diluted to 100 mL. The results were analyzed using GraphPad software.

## Western blot

SDS-PAGE of viral proteins was performed using the AmyKD-PAGE rapid gel preparation kit. Then, HAdV-1, HAdV-2, HAdV-3, HAdV-4, HAdV-5, HAdV-7, HAdV-21, and HAdV-55 were transferred through a semi-dry imprinting device onto a polyvinylidene fluoride membrane. The membrane was washed with 1× PBS containing 0.1% Tween 20 on a rocking bed three times for 5 min and sealed with 5% skimmed milk powder-PBST solution on a rocking bed slowly at room temperature for 2 h. After washing three times, the mAb (detecting antibody) was diluted with 2% skim milk powder-PBST solution to 2 µg/mL and incubated at 4℃ for 6–8 h. After washing, the goat anti-mouse IgG (H&L) HRP diluted to 1:5,000 with 2% skim milk powder-PBST solution was added, and the mixture was shaken slowly at room temperature for 1 h. Finally, ECL chemiluminescence was used to detect the specificity of different mAbs.

## Indirect immunofluorescence assay

A549 cells were cultured in T75 cell culture flasks. When the cells were in a logarithmic growth and good growth state, they were digested with trypsin and resuspended after centrifugation. The cells were inoculated into 24-well plates or confocal dishes at $1 \times 10^5$ cells/well, and after 24 h, the cell density reached 80%–90%. The medium was removed, and the viral dilutions of 10-fold dilution of HAdV-3 ($TCID_{50}/mL = 2.36 \times 10^5$) and HAdV-55 ($TCID_{50}/mL = 1.61 \times 10^5$) were added to each well at a volume of 300 µL/well and incubated for 2 h in 5% $CO_2$ at 37℃. Next, the viral dilution was sucked off, and 1 mL of maintenance medium was added into each well and incubated for 24 h in 5% $CO_2$ at 37℃. After 24 h, apparent cytopathic lesions were observed under a microscope, the medium was removed, and the cells were washed twice with PBS. Cells were fixed with 4% paraformaldehyde for 15 min at room temperature. After fixation, the cells were washed thrice with PBS for 5 min with vigorous shaking to remove paraformaldehyde. To ensure the antibody reached the antigen site, 0.5% Triton X-100 was added to permeabilize the cells for 15 min in the dark. After permeabilization, the cells were washed three times with PBS for 5 min. The cells were blocked with a solution prepared from 0.1% Triton X-100 and 10% goat serum for 30 min. After blocking, the cells were washed thrice with PBST for 5 min. The antibody solution diluted with PBS was added (the concentration of mAb was 4–5 µg/mL) and incubated at room temperature for 1 h or 4℃ overnight. After removing the primary antibody solution, the sections were washed thrice with PBST for 5 min. Fluorescein (FITC)-conjugated goat anti-mouse IgG (H + L) diluted in PBS was added at the recommended dilution ratio of 1:20–1:100; the cells were incubated at room temperature in the dark for 30 min. The secondary antibody solution was removed and washed thrice with PBST for 5 min. Hochest 33342 staining solution was added into each well containing 300-µL medium and incubated at room temperature in the dark for 5 min. Next, the cells were washed with PBS in the dark for 5 min. Finally, the anti-fluorescence quenching sealing agent was added dropwise at 50 µL/well. The images were captured using a fluorescence microscope. After capturing the images, fluorescence intensity analysis was performed using ImageJ software.

## Double-antibody sandwich ELISA

The immunoglobulin-specific capture antibody was diluted to a final concentration of 10 µg/mL in 1× PBS. The microtiter plate was coated by pipetting 100 µL of diluted antibody into individual wells and incubating for 2 h at 37℃. After washing, 200 µL of 5% (wt/vol) skimmed milk powder solution in 1× PBS was added to block the remaining protein-binding sites and incubated for 2 h at 37℃ or overnight at 4℃. After washing, the test antigen solutions for hexon protein, HAdV-3, and HAdV-55 were diluted 10 times, from initial concentrations of 10 µg/mL, $1.127 \times 10^4$ PFU/mL, and $4.13 \times 10^3$ PFU/mL in 1× PBS, respectively. Next, 100 µL of the dilutions was pipetted into each coated well and incubated for 30 min at 37℃, with monkeypox virus used as the negative control. After washing, the enzyme-labeled detection antibody was prepared, and direct

ELISA was used to obtain the optimal dilution titer. According to the manufacturer's instructions, using a glue-activated horseradish peroxidase labeling kit (no. MD010A), mAb 1F10 and mAb 8D3 were labeled by enzyme; 100 µL of the horseradish peroxidase (HRP)-labeled antibody was pipetted into each coated well and incubated for 30 min at 37°C. After washing, 100 µL of TMB substrate solution was added into each well, and the enzyme-based reaction was set for 10–30 min at 37°C in the dark. Finally, 50 µL of $H_2SO_4$ solution was added to each well to terminate the reaction. The results were analyzed using GraphPad software.

## mAb gene amplification

The hybridoma cells were cultured in a T75 culture flask to 70%–90% density, centrifuged to collect the cells, washed with 10 mL of 1× PBS, recentrifuged, washed repeatedly, and recentrifuged. RNA was extracted from the hybridoma cells using an AG Steady Pure Universal RNA Extraction Kit II (no. AG21022) and stored at –80°C. Second, the extracted RNA was reverse transcribed into cDNA, and the cDNA was stored at –20°C or –80°C using a Takara PrimeScript II 1st Strand cDNA Synthesis Kit (no. 6210A). Finally, the mAb was subjected to gene amplification of the light and heavy chains. Takara Taq Version 2.0 + dye premix enzyme (no. RR901A) was used for the PCR analysis of light and heavy chains. The PCR amplification protocol consisted of denaturation at 94°C for 5 min, denaturation at 94°C for 40 s, annealing at 58°C for 40 s, extension at 72°C for 1 min, a total of 35 cycles, and re-extension at 72°C for 10 min. The PCR products were analyzed using 1%–2% agarose gel electrophoresis, and the 300- to 500-bp bands were sequenced. The VBASE2 website (http://www.vbase2.org/) was used to analyze the sequencing results of the mAb and to obtain all variable region sequences.

## ACKNOWLEDGMENTS

This work was supported by grants from the National Natural Science Foundation of China (grant numbers: 32170939 and 82371846), the Guangdong Basic and Applied Basic Research Foundation (grant number 2022B1515020075), the Guangdong Science and Technology Program key projects (no. 2021B1212030014), and the Science and Technology Program of Guangzhou (202201011419).

The funding bodies had no role in the design of the study; the collection, analysis, and interpretation of data; or writing the manuscript.

We would like to thank Editage (www.editage.cn) for English language editing.

Conceptualization, S.C.G., X.W.W., Y.Y., Z.W., and W.C.S. Methodology, S.C.G., W.L.F., L.Y.H., Y.J.Z., Z.B., Y.C.B., J.Y.S., Z.L.L., W.Y.L., and W.Q.H. Investigation, W.L.F., Y.J.Z., L.Y.H., and Y.C.B. Writing—original draft, W.L.F. and L.Y.H. Writing—review and editing, S.C.G. and W.L.F. Funding acquisition, S.C.G., X.W.W., and Y.Y. Resources, S.C.G., Y.Y., Z.W., Z.B., X.W.W., and Z.L. Supervision, S.C.G. and X.W.W. Software, W.L.F. and L.Y.H.

The authors declare no conflicts of interest.

## AUTHOR AFFILIATIONS

[1]School of Public Health, Guangdong Medical University, Zhanjiang, China
[2]BSL-3 Laboratory (Guangdong), Guangdong Provincial Key Laboratory of Tropical Disease Research, School of Public Health, Southern Medical University, Guangzhou, China
[3]Department of Laboratory Medicine, Zhujiang Hospital, Southern Medical University, Guangzhou, China
[4]Shenzhen Key Laboratory of Pathogen and Immunity, State Key Discipline of Infectious Disease, Shenzhen Third People's Hospital, Second Hospital Affiliated to Southern University of Science and Technology, Shenzhen, China

## AUTHOR ORCIDs

Linfan Wu http://orcid.org/0009-0005-4687-9633
Wei Zhao http://orcid.org/0000-0003-1414-4438
Yang Yang http://orcid.org/0000-0002-6489-4459
Chenguang Shen http://orcid.org/0000-0001-8607-3750
Weiwei Xiao http://orcid.org/0009-0001-5759-5768

## FUNDING

| Funder | Grant(s) | Author(s) |
|---|---|---|
| MOST \| National Natural Science Foundation of China (NSFC) | 32170939, 82111530302 | Chenguang Shen |

## AUTHOR CONTRIBUTIONS

Linfan Wu, Data curation, Methodology, Writing – original draft | Yuhao Lin, Data curation, Methodology, Writing – original draft | Juzhen Yin, Investigation | Changbi Yang, Investigation | Yushan Jiang, Investigation | Linlin Zhai, Investigation | Yuelin Wang, Investigation | Li Zhu, Investigation | Qinghua Wu, Investigation | Bao Zhang, Methodology | Chengsong Wan, Investigation | Wei Zhao, Investigation | Yang Yang, Conceptualization | Chenguang Shen, Conceptualization, Funding acquisition, Writing – review and editing | Weiwei Xiao, Conceptualization

## ADDITIONAL FILES

The following material is available online.

### Supplemental Material

**Supplemental material (Spectrum01816-23-S0001.docx).** Fig. S1 to S4.

### Open Peer Review

**PEER REVIEW HISTORY (review-history.pdf).** An accounting of the reviewer comments and feedback.

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
