## [Reviewer comments · Microbiology Spectrum]

Microbiology Spectrum

Development of monoclonal antibodies targeting the conserved fragment of hexon protein to detect different serotypes of human adenovirus

Linfan Wu, Yuhao Lin, Juzhen Yin, Changbi Yang, Yushan Jiang, Linlin Zhai, Yuelin Wang, Li Zhu, Qinghua Wu, Bao Zhang, Chengsong Wan, Wei Zhao, Yang Yang, Chenguang Shen, and Weiwei Xiao

Corresponding Author(s): Chenguang Shen, Southern Medical University

Review Timeline:

Submission Date:	April 29, 2023
Editorial Decision:	September 3, 2023
Revision Received:	October 17, 2023
Editorial Decision:	December 2, 2023
Revision Received:	December 20, 2023
Accepted:	January 20, 2024

Editor: Jérôme Le Goff

Reviewer(s): Disclosure of reviewer identity is with reference to reviewer comments included in decision letter(s). The following individuals involved in review of your submission have agreed to reveal their identity: Arun Kumar Adhikary (Reviewer #1)

Transaction Report:

DOI: <https://doi.org/10.1128/spectrum.01816-23>

September 3, 2023

Dr. Chenguang Shen
Southern Medical University
Guangzhou
China

Re: Spectrum01816-23 (Development of monoclonal antibodies targeting the conserved fragment of Hexon protein for broadly detecting different serotypes of adenovirus)

Dear Dr. Chenguang Shen:

The manuscript requires extensive modifications before it can be considered for publication in Microbiology Spectrum. According to the reviewers' comments, there is room for improvement. In addition to significant scientific revisions, language editing is imperative.

Link Not Available

Sincerely,

Jérôme Le Goff

Journals Department
Reviewer comments:

Reviewer #1 (Comments for the Author):

The manuscript (Spectrum01816-23) entitled "Development of monoclonal antibodies targeting the conserved fragment of Hexon protein for broadly detecting different serotypes of adenovirus" by Linfan Wu et al. In their study, the authors developed monoclonal antibodies that can detect a few human adenoviruses. The development of diagnostic methods against human adenoviruses is important as there are many recombinant types included in the last few years. My queries regarding the manuscripts are the following:

Title: Include human adenovirus

Abstract:

The abstract does not match clearly with the experiment. Rewriting is necessary.

Introduction:

Use human adenovirus or HAdV throughout the manuscript

Lines 60, 63, 66 and others: Currently subgenus/subgroup of HAdV is designated as species. Use species throughout the manuscript.

Line 84: Write HAdV-2

Line 112: Write HAdV

Lines 118-123: The rationale for the development of monoclonal antibodies is not clear.

Is there any targeted antiviral against HAdVs? If any, please mention.

Materials and method:

Lines 148-150: Write sequences of which types were obtained from the Genbank with their accession number.

Use more types of respiratory HAdVs for alignment as mentioned in the introduction and make a figure of alignment of the include conserved region.

Lines 148-160: provide web site address of all the editing tools.

Mention why you selected core protein HAdV IVa2 in this study.

Line 187: Write BALB/c mice

Result:

Line 285-91:

HAdV-2, HAdV-4, HAdV-5, 286 HAdV-7, HAdV-11 and HAdV-21 are included for alignment and phylogenesis. Currently, HAdV-3 is a mostly recovered respiratory isolate. It seems HAdV-3 is not included in alignment or phylogenesis. Use all types of respiratory HAdVs (including 3) for alignment as mentioned in the introduction and make a figure of alignment that include conserved regions.

Discussion:

HAdV-3, HAdV-4, HAdV-5, HAdV-7 and HAdV-55 are used to check the binding activity of mAbs. I am not sure why other types are not tested.

Overall, the writing of the manuscript is not clear. There are many minor mistakes The experimental part also needed to revise. The authors can use professional editing services for improvement.

Reviewer #2 (Comments for the Author):

This paper details the generation of cross-reactive mAbs against specific epitopes within HAdV proteins hexon and IVa2. It is more of a technical report, with little science in terms of hypothesis etc. The mAbs generated could perhaps be useful for HAdV diagnostics but that wasn't directly tested or benchmarked against more sensitive techniques like qPCR, and only a small subset of HAdV types were tested via western blot and ELISA.

Reviewer #3 (Comments for the Author):

In the current manuscript, Wu-Lin-Yin et al. have described the production of 6 mAbs, out of which 3 mAbs bind specifically to Hexon protein on the surface of AdV particles. The data presented appears robust and the applicability of broad-spectrum hexon antibody would be great. I have a few specific suggestions experimentally mentioned below.

1. The authors need to clarify if the hexon mAbs are functional under native or denaturing conditions in western blots. This will give hints as to where these antibodies bind. They should also check in immunofluorescent whether the antibody works before or after AdV replication. To add to that, one of the important uses of highly specific hexon antibodies (e.g., 9C12) is their ability to be used during virus entry studies. The authors should hence test in parallel with 9C12 whether incoming AdVs can be detected with either of these antibodies. In any case, 9C12 antibody should be included as a comparison for hexon mAbs.
2. Can any of the 3 hexon antibodies promote AdV neutralisation? This simple experiment (i.e., pre-incubating the antibodies with virions before binding to cells etc.) would clarify if any of these bind to the surface exposed regions.
3. The manuscript can be improved in terms of the language of the paper which needs a strong reevaluation and sentences in several places need to be appended.
4. Include statistical tests in Figs 4 and 5.

Staff Comments:

Preparing Revision Guidelines

Please return the manuscript within 60 days; if you cannot complete the modification within this time period, please contact me. If you do not wish to modify the manuscript and prefer to submit it to another journal, please notify me of your decision immediately so that the manuscript may be formally withdrawn from consideration by Microbiology Spectrum.

The manuscript (Spectrum01816-23) entitled "Development of monoclonal antibodies targeting the conserved fragment of Hexon protein for broadly detecting different serotypes of adenovirus" by Linfan Wu et al. In their study, the authors developed monoclonal antibodies that can detect a few human adenoviruses. The development of diagnostic methods against human adenoviruses is important as there are many recombinant types included in the last few years. My queries regarding the manuscripts are the following:

Title: Include human adenovirus

Abstract:

The abstract does not match clearly with the experiment. Rewriting is necessary.

Introduction:

Use human adenovirus or HAdV throughout the manuscript

Lines 60, 63, 66 and others: Currently subgenus/subgroup of HAdV is designated as species. Use species throughout the manuscript.

Line 84: Write HAdV-2

Line 112: Write HAdV

Lines 118-123: The rationale for the development of monoclonal antibodies is not clear.

Is there any targeted antiviral against HAdVs? If any, please mention.

Materials and method:

Lines 148-150: Write sequences of which types were obtained from the Genbank with their accession number.

Use more types of respiratory HAdVs for alignment as mentioned in the introduction and make a figure of alignment of the include conserved region.

Lines 148-160: provide web site address of all the editing tools.

Mention why you selected core protein HAdV IVa2 in this study.

Line 187: Write BALB/c mice

Result:

Line 285-91:

HAdV-2, HAdV-4, HAdV-5, 286 HAdV-7, HAdV-11 and HAdV-21 are included for alignment and phylogenesis. Currently, HAdV-3 is a mostly recovered respiratory isolate. It seems HAdV-3 is not included in alignment or phylogenesis. Use all types of respiratory HAdVs (including 3) for alignment as mentioned in the introduction and make a figure of alignment that include conserved regions.

Discussion:

HAdV-3, HAdV-4, HAdV-5, HAdV-7 and HAdV-55 are used to check the binding activity of mAbs. I am not sure why other types are not tested.

Overall, the writing of the manuscript is not clear. There are many minor mistakes The experimental part also needed to revise. The authors can use professional editing services for improvement.

Dear editor Jérôme Le Goff and dear reviewers,

Re: Manuscript ID: Spectrum01816-23 and Title: Development of monoclonal antibodies targeting the conserved fragment of Hexon protein for broadly detecting different serotypes of human adenovirus.

Editor Comments:

The manuscript requires extensive modifications before it can be considered for publication in Microbiology Spectrum. According to the reviewers' comments, there is room for improvement. In addition to significant scientific revisions, language editing is imperative.

Response: Thank you very much for your comments and professional advice. These opinions help to improve the academic rigor of our article. Based on your suggestion and request, we have corrected the revised manuscript's modifications. Meanwhile, we have carefully polished the language of our manuscript as suggested. It is hoped that this version of the article will be accepted by the Microbiology Spectrum

Reviewer(s)' Comments to Author:

Reviewer 1#

Comments for the Author

The manuscript (Spectrum01816-23) entitled "Development of monoclonal antibodies targeting the conserved fragment of Hexon protein for broadly detecting different serotypes of adenovirus" by Linfan Wu et al. In their study, the authors developed monoclonal antibodies that can detect a few human adenoviruses. The development of diagnostic methods against human adenoviruses is important as there are many recombinant types included in the last few years. My queries regarding the manuscripts are the following:

Response: Thanks for your time and the insightful comments about our study.

1. Title: Include human adenovirus ; Abstract: The abstract does not match clearly with the experiment. Rewriting is necessary.

Response: Thanks for your time and comments on our manuscript, we have revised the title, and the abstract has been rewritten (lines 2-40).

2. Introduction:

Use human adenovirus or HAdV throughout the manuscript.

Lines 60, 63, 66 and others: Currently subgenus/subgroup of HAdV is designated as species. Use species throughout the manuscript.

Line 84: Write HAdV-2

Line 112: Write HAdV

Lines 118-123: The rationale for the development of monoclonal antibodies is not clear. Is there any targeted antiviral against HAdVs? If any, please mention.

Response: Thank you for your scrutiny and suggestions. We are very sorry for our carelessness. We have changed the adenovirus in the manuscript to human adenovirus

or HAdV and subgenus changed to species (lines 65-75). Then, we have corrected the correction of line 84 (lines 88-89) and line 112 (line 120). At the same time, we have added the basic principle of monoclonal antibody development (lines 126-136). In addition, we answered question about anti-human adenovirus drugs (lines 139-144).

3. Materials and method:

Lines 148-150: Write sequences of which types were obtained from the GenBank with their accession number.

Use more types of respiratory HAdVs for alignment as mentioned in the introduction and make a figure of alignment of the include conserved region.

Response: Thanks for your insightful comments and suggestions. We added more human adenovirus types and their accession numbers from GenBank, as shown in Table 1 (lines 199-200). Meanwhile, we have drawn a sequence alignment analysis diagram containing conserved regions, as shown in Figure.1 of the results.

4. Lines 148-160: provide web site address of all the editing tools. Mention why you selected core protein HAdV IVa2 in this study. Line 187: Write BALB/c mice.

Response: Thanks for your insightful comments and suggestions. In this manuscript, we have edited the website address of all the tools used (lines 185, 188, 195-196, 374). In the previous work, we compared all human adenovirus sequences. Finally, we selected the three most conserved antigens, one of which is IVa2 (lines 192-193), and as shown in Figure.1 of the results. Furthermore, we've supplemented all the BALB/c mice, as shown in supplementary Fig. 2 of the Supplementary Materials.

5. Result:

Line 285-291:

HAdV-2, HAdV-4, HAdV-5, HAdV-7, HAdV-11 and HAdV-21 are included for alignment and phylogenesis. Currently, HAdV-3 is a mostly recovered respiratory isolate. It seems HAdV-3 is not included in alignment or phylogenesis. Use all types of respiratory HAdVs (including 3) for alignment as mentioned in the introduction and make a figure of alignment that include conserved regions.

Response: Thanks for your insightful comments and suggestions. We are sorry for our negligence. We have supplemented the homology analysis and conserved region alignment of HAdV-3, HAdV-1 and HAdV-55 (lines 379-382), as shown in Figure.1 of the results.

6. Discussion:

HAdV-3, HAdV-4, HAdV-5, HAdV-7 and HAdV-55 are used to check the binding activity of mAbs. I am not sure why other types are not tested.

Overall, the writing of the manuscript is not clear. There are many minor mistakes. The experimental part also needed to revise. The authors can use professional editing services for improvement.

Response: Thanks for your insightful comments and suggestions. We added the specific detection of monoclonal antibodies and HAdV-1, HAdV-2, HAdV-21 by

Western blot (lines 473-476). At present, HAdV-1, HAdV-2, HAdV-3, HAdV-4, HAdV-5, HAdV-7, HAdV-21 and HAdV-55 have been successfully isolated and cultured in the Third People's Hospital of Shenzhen, Guangdong Province. Therefore, we only used the above 8 types of human respiratory adenovirus detection. At last, we tried our best to improve the manuscript and polish the language of our manuscript. We appreciate for your warm work earnestly.

Reviewer 2#

Comments for the Author

1.This paper details the generation of cross-reactive mAbs against specific epitopes within HAdV proteins Hexon and IVa2. It is more of a technical report, with little science in terms of hypothesis etc.

Response: We sincerely thank you for your careful reading. In our work, by analyzing and comparing the gene sequences of all human adenoviruses, the three most conserved antigen fragments were selected. By expressing the antigens, immunizing mice, preparing mouse-derived monoclonal antibodies and identifying the specificity of monoclonal antibodies to HAdVs. In this process, our hypothesis is relatively small. However, the monoclonal antibodies we discovered are innovative and have certain scientific value, and the discovery of these monoclonal antibodies is conducive to the diagnosis and treatment of human adenovirus.

2.The mAbs generated could perhaps be useful for HAdV diagnostics but that wasn't directly tested or benchmarked against more sensitive techniques like qPCR, and only a small subset of HAdV types were tested via western blot and ELISA.

Response: Thanks for your insightful comments and suggestions. We also added the specific detection of monoclonal antibodies and HAdV-1, HAdV-2, HAdV-21 by Western blot. A total of 8 types of human respiratory adenovirus were detected (lines 678-680), as shown Figure.7 of the results (lines 696). In addition, we supplemented the double antibody sandwich ELISA test (lines 426-444). At the same time, qPCR was used to detect human adenovirus, and the results were compared with those of double antibody sandwich ELISA (lines 734-763). Finally, we appreciate your suggestion earnestly.

Reviewer 3#

Comments for the Author

In the current manuscript, Wu-Lin-Yin et al. have described the production of 6 mAbs, out of which 3 mAbs bind specifically to Hexon protein on the surface of AdV particles. The data presented appears robust and the applicability of broad-spectrum Hexon antibody would be great. I have a few specific suggestions experimentally mentioned below.

Response: Thanks for your time and the insightful comments about our study.

1. The authors need to clarify if the Hexon mAbs are functional under native or

denaturing conditions in western blots. This will give hints as to where these antibodies bind. They should also check in immunofluorescent whether the antibody works before or after AdV replication. To add to that, one of the important uses of highly specific Hexon antibodies (e.g., 9C12) is their ability to be used during virus entry studies. The authors should hence test in parallel with 9C12 whether incoming AdVs can be detected with either of these antibodies. In any case, 9C12 antibody should be included as a comparison for Hexon mAbs.

Response: Thanks for your insightful comments and suggestions. By Western blotting, we detected the specificity of monoclonal antibodies against human adenovirus antigens under natural or denatured conditions. As previously reported, monoclonal antibodies could only detect human adenovirus antigens under denatured conditions by Western bolt (lines 482-485). As suggested by the reviewer, we have supplemented the indirect immunofluorescence assay (lines 308-338). The results suggest that monoclonal antibodies work after HAdV replication (lines 500-509). We have reviewed many articles on highly specific Hexon antibodies, but we have not found the relevant gene sequences, and couldn't express and compare the corresponding mouse antibodies (lines 584-587). However, we detected human adenovirus by highly sensitive qPCR and compared it with our results, as shown in Figure 6 of the results.

2. Can any of the 3 Hexon antibodies promote AdV neutralization? This simple experiment (i.e., pre-incubating the antibodies with virions before binding to cells etc.) would clarify if any of these binds to the surface exposed regions.

Response: Thanks for your insightful comments and suggestions. We performed neutralization tests on these three monoclonal antibodies but failed to show satisfactory results (lines 572-574).

3. The manuscript can be improved in terms of the language of the paper which needs a strong reevaluation and sentences in several places need to be appended.

Response: Thanks for your insightful comments and suggestions. We have tried our best to modify the details and polish the language of the manuscript.

4. Include statistical tests in Figs 4 and 5.

Response: Thanks for your insightful comments and suggestions. Figure 4 and Figure 5 in the original manuscript are now Figure 2 and 3, respectively. We have passed the statistical tests and replotted, as shown in Figure 2 and Figure 3 of the results.

Thank you very much for your attention and time. Look forward to hearing from you.

Re: Spectrum01816-23R1 (Development of monoclonal antibodies targeting the conserved fragment of Hexon protein for broadly detecting different serotypes of human adenovirus)

Dear Dr. Chenguang Shen:

Thank you for the privilege of reviewing your work. Below you will find my comments, instructions from the Spectrum editorial office, and the reviewer comments.

We appreciate the authors for their responses to reviewers' comments aimed at improving the manuscript. The document requires revision by a native speaker specializing in scientific article editing.

Additionally, the introduction, discussion, and conclusion need modification. While specific HAdV antibodies can confirm that individuals have been infected by HAdV, they cannot be used to confirm an ongoing infection and cannot establish a direct link with symptoms. Since the antibodies developed in this study are non-type-specific, they cannot be employed to determine the HAdV type by which an individual could be infected. Such antibodies could be of interest in describing the natural history of HAdV infection and understanding how HAdV concentrations in body fluids correlate with protection or the severity of infection

Revision Guidelines

Sincerely,
Jérôme Le Goff
Editor
Microbiology Spectrum

Dear editor Jérôme Le Goff,

Re: Manuscript ID: Spectrum01816-23R1 and Title: Development of monoclonal antibodies targeting the conserved fragment of Hexon protein for broadly detecting different serotypes of human adenovirus.

Editor Comments:

We appreciate the authors for their responses to reviewers' comments aimed at improving the manuscript. The document requires revision by a native speaker specializing in scientific article editing.

Additionally, the introduction, discussion, and conclusion need modification. While specific HAdV antibodies can confirm that individuals have been infected by HAdV, they cannot be used to confirm an ongoing infection and cannot establish a direct link with symptoms. Since the antibodies developed in this study are non-type-specific, they cannot be employed to determine the HAdV type by which an individual could be infected. Such antibodies could be of interest in describing the natural history of HAdV infection and understanding how HAdV concentrations in body fluids correlate with protection or the severity of infection.

Response: Thank you very much for your comments and professional advice. These opinions help to improve academic rigor of our article. Based on your suggestion and request, We have revised the article and all the changes have been highlighted in red. According to your advice, this manuscript was edited for proper English language, grammar, punctuation, spelling, and overall style by one or more of the highly qualified native English speaking editors at Editage (www.editage.cn). Meanwhile, we have modified the introduction, discussion, and conclusion accordingly (lines 174-177, 351-355, and 364-368).

Thank you again for your valuable comments and suggestions. Look forward to hearing from you.

Re: Spectrum01816-23R2 (Development of monoclonal antibodies targeting the conserved fragment of hexon protein to detect different serotypes of human adenovirus)

Dear Dr. Chenguang Shen:

Please make the following modifications:

Abstract. Change the conclusion as follows : The monoclonal antibodies developed in this study show potential for detecting human Adenoviruses.

Introduction. Please remove the statement regarding cell culture as the gold standard, as molecular biology now takes precedence.

Your manuscript has been accepted, and I am forwarding it to the ASM production staff for publication. Your paper will first be checked to make sure all elements meet the technical requirements. ASM staff will contact you if anything needs to be revised before copyediting and production can begin. Otherwise, you will be notified when your proofs are ready to be viewed.

Sincerely,
Jérôme Le Goff
Editor
Microbiology Spectrum